# Projection of Forest Fire Danger due to Climate Change in the French Mediterranean Region

**Vassiliki Varela** [1,*]**, Diamando Vlachogiannis** [1,*]**, Athanasios Sfetsos** [1] 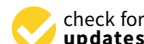**, Stelios Karozis** [1]**, Nadia Politi** [1] **and Frédérique Giroud** [2]

1    Environmental Research Laboratory, INRASTES, NCSR "Demokritos", 15341 Agia-Paraskevi, Greece
2    Département essais et recherche, CEREN/ENTENTE, 13120 Gardanne, Provence-Alpes-Côte d'Azur, France
\*    Correspondence: vvarela@ipta.demokritos.gr (V.V.); mandy@ipta.demokritos.gr (D.V.)

**Abstract:** Fire occurrence and behaviour in Mediterranean-type ecosystems strongly depend on the air temperature and wind conditions, the amount of fuel load and the drought conditions that drastically increase flammability, particularly during the summer period. In order to study the fire danger due to climate change for these ecosystems, the meteorologically based Fire Weather Index (FWI) can be used. The Fire Weather Index (FWI) system, which is part of the Canadian Forest Fire Danger Rating System (CFFDRS), has been validated and recognized worldwide as one of the most trusted and important indicators for meteorological fire danger mapping. A number of FWI system components (Fire Weather Index, Drought Code, Initial Spread Index and Fire Severity Rating) were estimated and analysed in the current study for the Mediterranean area of France. Daily raster-based data-sets for the fire seasons (1st May–31st October) of a historic and a future time period were created for the study area based on representative concentration pathway (RCP) 4.5 and RCP 8.5 scenarios, outputs of CNRM-SMHI and MPI-SMHI climate models. GIS spatial analyses were applied on the series of the derived daily raster maps in order to provide a number of output maps for the study area. The results portray various levels of changes in fire danger, in the near future, according to the examined indices. Number of days with high and very high FWI values were found to be doubled compared to the historical period, in particular in areas of the Provence-Alpes-Côte d'Azur (PACA) region and Corsica. The areas with high Initial Spread Index and Seasonal Spread Index values increased as well, forming compact zones of high fire danger in the southern part of the study area, while the Drought Code index did not show remarkable changes. The current study on the evolution of spatial and temporal distribution of forest fire danger due to climate change can provide important knowledge to the decision support process for prevention and management policies of forest fires both at a national and EU level.

**Keywords:** fire danger; climate change; RCP; FWI; extreme weather; drought index

---

## 1. Introduction

Climate change will result in future altered fire regimes, which will be realized through changes in fire weather, fire behaviour and carbon emissions, according to several research works [1].

Many areas across the world have seen a rise in extreme fires in recent years. Those include western U.S. states and southern and western Europe. They also include unexpected places above the Arctic Circle, like the fires in Sweden during the summer of 2018 [2,3].

Extreme fire events, which are also referred to as "megafires", are becoming frequent on a global scale; recent fires in Portugal, Greece, California and other areas confirm this fact. There is not complete agreement on the term "megafires", which often refers to catastrophic fire events in terms of human casualties, economic losses or both. Ayanz et al. [4], analysed some of the most catastrophic fire episodes

in Europe and related the events to existing conditions in terms of number of fires and burnt areas in the countries and regions where they occurred, showing that these extreme events did not follow the general trend of fire history in the area but constituted outstanding critical events characterized on the basis of the meteorological and fire danger conditions prior to and during development of the event. More particularly, those extreme events were driven by critical weather conditions that led to a concentration of numerous large fires in time and space (fire clusters). The simultaneity in fire ignitions and rapid fire spread prevented efficient initial fire attacks.

According to Vose et al. [5], who conducted research in the United States, by the end of the 21st century, rapidly visible short-term effects on forest ecosystems will be caused by altered disturbance regimes, including wildfire increments, which will also affect significantly the vulnerability of many areas, since both infrastructure and resource production are based on current climate and steady-state conditions.

Climate change will reduce fuel moisture levels from present values around the Mediterranean region and the region will become drier, increasing the weather-driven danger of forest fires. The countries in highest danger are Spain, Portugal, Turkey, Greece, parts of central and southern Italy, Mediterranean France and the coastal region of the Balkans, according to recent research of the Joint Research Centre (JRC) [6]. Recognizing the need to minimize the adverse effects of climate change in such fire-prone areas must drive the efforts of the fire management and prevention community towards detailed assessment of future impacts. In view of that, the aim of this work was to provide a detailed study of the fire danger evolution and expected changes in the fire regime of a region of South France in the near future, using Fire Weather Index (FWI) system indices, calculated using two available climate model datasets for two representative concentration pathway (RCP) scenarios (RCP 4.5 and RCP 8.5 of the Intergovernmental Panel on Climate Change (IPCC)).

The Fire Weather Index FWI, which is part of the Canadian Forest Fire Danger Rating System (CFFDRS), has been validated and recognized worldwide as one of the most trusted and important indicators for meteorological fire danger mapping.

FWI is used for daily fire danger mapping, although it has also been used with projected climatic data for investigating future changes in the frequency of occurrence and intensity of natural disasters [7–9]. According to the studies of Cloppet et al. [7], Météo-France found that there was a clear statistical link between FWI and fire frequency and size in France, and the results showed an increase in fire meteorological danger over the last 50 years with strong regional trends. In those works, Météo-France also studied the climate change potential impact on forest fire risk with climate change scenarios (ARPEGE V4 model with 3 emissions scenarios: A1B, A2 and B1) and showed that there is a clear trend of increase of total number of days of FWI above a threshold. According to this study, a huge increase in forest fire risk is expected by 2060.

## 2. Materials and Methods

### 2.1. Calculation and Classification of FWI System Components

The Fire Weather Index (FWI) system is one of the two major subsystems of the Canadian Forest Fire Danger Rating System (CFFDRS) [10,11]. FWI system components are calculated with a rather complicated algorithm. Van Wagner [11] describes extensively the structure of the FWI System and its components. The algorithm and the equations using a Fortran program are also described in detail by Van Wagner et al. [12].

The FWI system was initially applied in Canada in the early 1970s [10,11] and later in other regions such as the Mediterranean [13,14]. In particular, Viegas et al. [13] reported that in a comparative study for fire danger evaluation methods using real data in France, Italy and Portugal, the FWI system showed the best overall performance in spite of the fact that the climate and vegetation of the Mediterranean region differ to those of Canada. At present, fire risk maps are produced at the European level by the Joint Research Centre (JRC) using the FWI system [15,16].

The FWI system has also been used for studying climate change effects in extreme fire weather and fire regimes in several areas of the world [17–20].

The FWI system is comprised of six components: three fuel moisture codes, which are first level intermediate outputs of the system (i.e., Fine Fuel Moisture Code, Duff Moisture Code, Drought Code) two fire behaviour indices, which are second level intermediate outputs (i.e., Initial Spread Index, BuildUp Index) and the final output, the Fire Weather Index (Figure 1). The Daily Severity Rating (DSR) and its time-averaged value, the Seasonal Severity Rating (SSR), are extensions of the FWI System. The DSR is a transformation of the daily FWI value, calculated as follows [10]:

$$DSR = 0.0272FWI^{1.77} \tag{1}$$

Higher FWI values are emphasized through the power relation. The DSR can be accumulated over time as the cumulative DSR, or it may be averaged over time as the SSR [10]. The context of the individual FWI system components which were selected in this study is described briefly below:

- The Drought Code (DC) is a numeric rating of the average moisture content of deep, compact organic layers. This code is a useful indicator of seasonal drought effects on forest fuels and the amount of smouldering in deep duff layers and large logs.
- The Initial Spread Index (ISI) is a numeric rating of the expected rate of fire spread. It combines the effects of wind and the Fine Fuel Moisture Code on rate of spread without the influence of variable quantities of fuel.
- FWI represents the potential fireline intensity and it is a good indicator of general fire danger [13].
- The Daily Severity Rating (DSR) is a numeric rating of the difficulty of controlling fires. It is based on the Fire Weather Index but more accurately reflects the fire suppression expected efforts [11]. SSR is the mean value of the DSR during a fire season.

Daily meteorological values at noon (12:00 h) of near surface temperature, relative humidity and 10-m wind speed as well as 24-h cumulative precipitation are used for the calculation of the components of the system. The values of FWI are usually found in the range of 0 to 100. Values above 100 can be calculated for more extreme meteorological conditions. For operational purposes and depending on the application area, the range of FWI values can be categorised into 4 to 6 classes that denote different fire danger level [21–24].

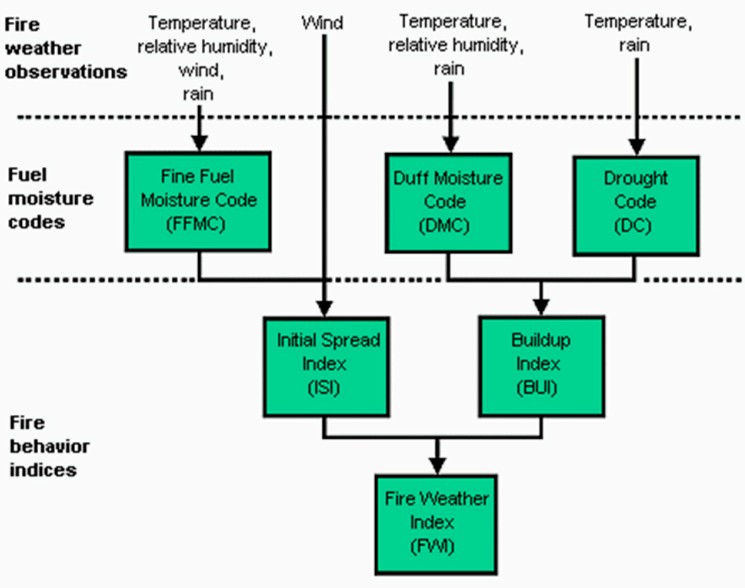

**Figure 1.** Structure components and flow diagram of FWI calculation process [10,11].

## 2.2. Description of the Study Area

The study area is delineated in the north by the Alps, south by the Mediterranean shore, east by the Italian Border and west by the Spanish border, and it includes also the island of Corsica (Figure 2a).

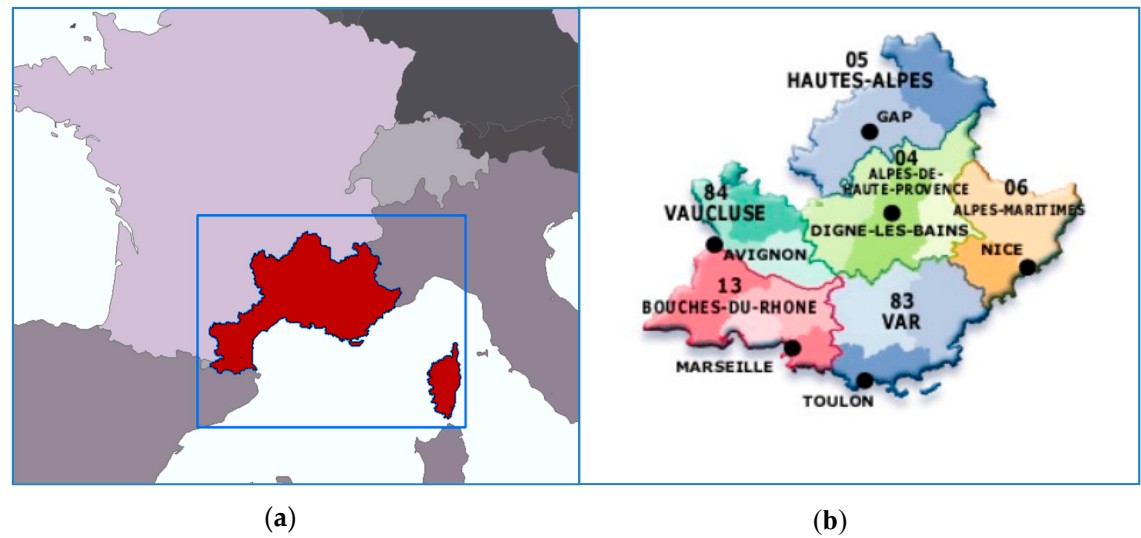

|     |     |
| :-: | :-: |
| (**a**) | (**b**) |

**Figure 2.** The study area: (**a**) Overview of the study area; (**b**) subareas of the Provence-Alpes-Côte d'Azur (PACA) region.

The study area is an area of about 34,000 km$^2$ with a population of over 6 million inhabitants and includes the Provence-Alpes-Côte d'Azur (PACA) region and island of Corsica (Figure 2b). Three main geographical zones can be distinguished in areas that have different physiographic features. The Mediterranean coast is characterized by low shores and cliffs and has strong socioeconomic dynamics, especially in the Bouches-du-Rhône region. The natural reserve of Camargue is an exception, where human activities are quite limited. Alluvial plains, lowlands and hills are the main features of the central and the western parts, while the dense traffic network of the Rhône and Durance Valleys supports significant economic development in this region. On the contrary, the Provence hinterland and the Alpine range (located north–east of the Region) has a sparse population and the abandonment of agricultural activities enabled reforestation and the expansion of a wildland–urban interface (WUI) zone.

The whole area is characterized by a Mediterranean climate with mild and humid winters and a lot of sunshine, warm and dry summers and heavy and sudden precipitation events and strong winds (Mistral, Tramontane, Lombarde). These conditions are favourable to the occurrence and spread of forest fires every year during the fire-season [25–27]. Within the area of study, two climate types can be identified, the coastal warm and sunny type and the inland mountainous with harsh winters.

Heat wave induced forest fires are a frequent climate hazard and severe heat wave conditions, so-called "canicule", are the cause of hard to control and high impact forest fires. "Canicule" (i.e., severe heat wave) is officially defined as follows: during at least 3 full days, the minimum temperature (at night) is not lower than 24 °C (Bouches-du-Rhône and Alpes Maritimes) or 23 °C (Var); the maximum temperature exceeds 35 °C (Bouches-du-Rhône and Var) or 31 °C (Alpes–Maritimes).

## 2.3. Climate Models and RCP Scenarios

EURO-CORDEX—Coordinated Downscaling Experiment, European Domain [28]—created by the Task Force for Regional Climate Downscaling of the World Climate Research Program, is an internationally coordinated effort to deliver European climate predictions for the 21$^{st}$ century at a regional scale at ~50 km (0.44°) and ~12.5 km (0.11°) horizontal resolution [29]. In EURO-CORDEX, a rather large ensemble of model simulations of the European domain has been carried out for

historic climate and future climate predictions based on Coupled Model Intercomparison Project (CMIP5) Global Climate Models (GCM) projections. The EURO-CORDEX climate projections have been simulated considering the three RCP greenhouse gas and pollutant emission scenarios (RCP 2.6, RCP 4.5 and RCP 8.5) [30]. EURO-CORDEX datasets emanate from a coordinated model evaluation framework and are regarded today as the most comprehensive and representative climate projections for the region for climate change impact assessment studies [31].

In the current study, regional climate simulations of high spatial resolution (0.11°, ~12.5 km) from the EURO-CORDEX database were used to investigate the impact of climate change on fire danger according to RCP 4.5 and RCP 8.5. Details for the development of the RCP emissions scenarios are provided in Moss et al. [29] and van Vuuren et al. [32]. Each RCP scenario represents a different pathway in space and time throughout the 21st century of the radiative forcing due to varying emissions of human activities and land use changes that result in a certain value of it at the end of the century. The RCP 8.5—business as usual scenario—is the most severe in which emissions are assumed to rise throughout the century causing a value of 8.5 Watts $m^{-2}$ in the radiative forcing in 2100, and it is regarded as the most credible prediction if no mitigation actions are taken. Emissions according to RCP 4.5 peak around 2040 at around 50% higher than 2000 levels, declining rapidly over 30 years and stabilising at half of 2000 levels thereafter until the end of the century, when the radiative forcing attains a value of 4.5 Watts $m^{-2}$. These are the reasons why the two RCP scenarios were selected, as RCP 4.5 peaks within the future period investigated (2036–2045) and RCP 8.5 as a worst case scenario is widely simulated in impact assessment studies [33].

Output data from the EURO-CORDEX regional climate model RCA4 of the Rossby Centre [34], were available from the openly accessible SMHI (Swedish Meteorological and Hydrological Institute) archives on NetCDF format. More specifically, the climate projections of RCA4 for the two RCP scenarios and from two forcing Global Circulation Models (GCMs), those of CNRM-CERFACS-CNRM-CM5 [35] and MPI-M-MPI-ESM-LR [36], were used. The climate RCA4 model data consist of atmospheric variables, such as temperature, relative humidity, wind speed and wind direction at 12:00 hr as well as total 24 h precipitation at 12.5 km spatial resolution. The final datasets of the above mentioned variables used for the study were processed at 9 km spatial resolution for the domain of interest after applying regridding based on the nearest neighbour interpolating process. The RCA4 model performance in the recent past climate was evaluated by Strandberg et al. [37]. In this work, the aim was to delineate the approach for producing projections of fire danger in a region employing climate projection data of atmospheric variables in the FWI system.

*2.4. Methodology of Climatic Analysis and Mapping of FWI System Parameters*

The projections of critical meteorological parameters that are required for the calculation of FWI system parameters in the future were calculated with the use of the data from the two climate models and the two climate scenarios described above.

The atmospheric climate data sets that were elaborated for the area under consideration were used for the calculation of daily parameter values of the FWI system for the fire seasons (1st of May to 31st of October), of a future time period of ten years (2036–2045) for the two climate models and the two RCP scenarios. Thus, four data series were created corresponding to each climate model and scenario. As there can be discernible differences between the outputs of different climate models, the representation for a variable in a study area can be improved by taking the mean value of the variable from such climate datasets [28]. This approach was followed in the current study, where the mean value of the atmospheric variable from the two available regional climate datasets was calculated at each grid point. Hence, each RCP scenario was represented by the mean value of the variable (temperature, precipitation, wind speed and direction).

An additional "historic period" data series was considered representing the reference time period, based on the climate model MPI-SMHI and RCP 4.5 for the fire seasons of the years 2006–2015.

Here it must be emphasised that all system components as depicted in Figure 1 were calculated for the current study, as it was clear (as seen in Figure 1) that the intermediate (first level and second level) system components were strongly interdependent. However, only the most important system components based on their context for the fire danger evaluation and the purposes of this study are discussed and shown in figures. Thus, the following parameters were selected:

a.　Drought Code (DC) as one of the most important of the first level intermediate outputs of the FWI system;
b.　Initial Spread Index (ISI) as one of the second level intermediate outputs of the FWI system;
c.　Fire Weather Index (FWI), which was the final output;
d.　Seasonal Severity Rating (SSR) as the parameter expressing fire danger for an entire fire season.

Depending on the parameter, mean and maximum values as well as total number of days of high and extreme values for the historic and future periods were calculated for each spatial unit (cell) of the area of interest. Corresponding raster map layers were created for further spatial analyses and evaluation of the results. More particularly, the spatial datasets that were created and analysed were:

(a)　Seasonal Severity Rating (SSR) for the historic and future fire period for the climatic scenarios RCP 4.5 and RCP 8.5;
(b)　Spatial future changes of SSR for RCP 4.5 and RCP 8.5;
(c)　Maximum values of Initial Spread Index for the historic and future 10-year fire periods for RCP 4.5 and RCP 8.5;
(d)　Drought Code mean values for the historic and future 10-year fire periods for RCP 4.5 and RCP 8.5;
(e)　Number of days per fire period for the historic and future 10-year fire periods for RCP 4.5 and RCP 8.5.

In addition, statistical elaboration of the resulting spatial datasets (i.e., maps) was performed for FWI, ISI and DC indices, for calculating how the total area of interest, the size of which was 34,290 km$^2$, was distributed in the categories of values for each index, both in km$^2$ and in percentages of the total area.

## 3. Results

The results of the spatial analyses for the above mentioned FWI system parameters are presented below.

### 3.1. Seasonal Severity Rating Mapping

Figure 3 shows the Seasonal Severity Rating maps for the historic period (i.e., fire periods of the years 2006–2015) and for the future period (fire period of the years 2036–2045) for two climatic RCP scenarios (RCP 4.5, RCP 8.5). According to these results, there was a significant change in SSR values in both RCP scenarios in the examined future period for the whole study area. The southern part of the area was mostly affected where zones with SSR values greater than ten appeared, while in the historic period no areas with SSR values greater than eight occurred.

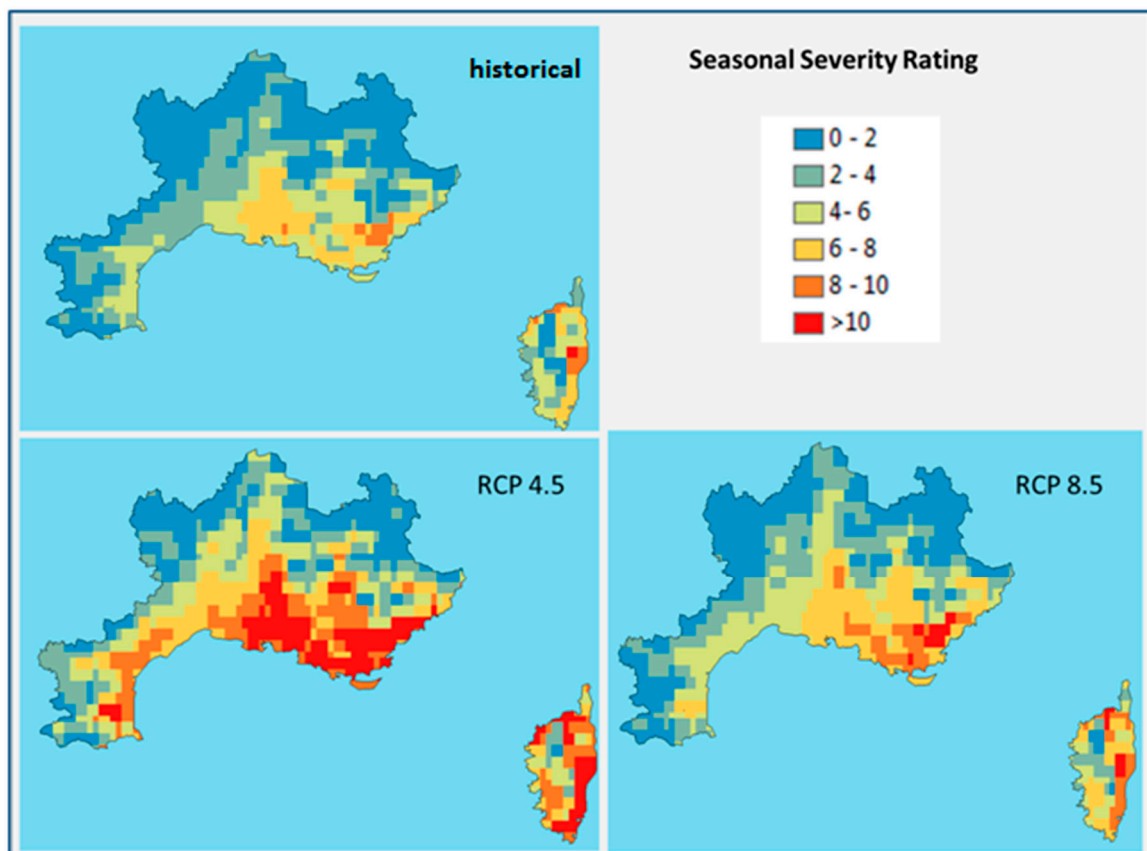

**Figure 3.** Maps of the historic (2006–2015) and near future (2036–2045) Seasonal Severity Rating parameter, for RCP 4.5 (left map) and RCP 8.5 (right map).

The changes of SSR in the future period for both RCP scenarios are shown more clearly in the maps in Figure 4, where the differences of the SSR values between the RCP scenarios and the historic data were estimated. As is presented in these maps, small (up to 2) to significant (4 to 7 units) changes occurred in almost all the study area. RCP 4.5 appeared as the most serious one for this variable, especially for the areas near the Mediterranean coast and Corsica.

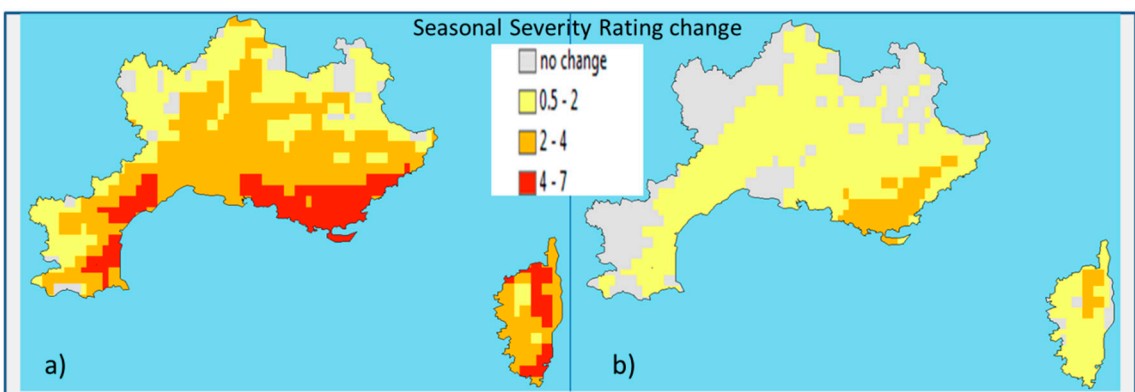

**Figure 4.** Spatial distribution of the Seasonal Severity Rating differences in the near future for: (**a**) RCP4.5; (**b**) RCP8.5.

### 3.2. Initial Spread Index Mapping

The expected near future changes in maximum Initial Spread Index were quite remarkable (Figure 5, Table 1). As seen in Table 1 there was an approximately 10% increase of the area with

ISImax in the three higher categories of values (i.e., ISImax > 45) in both RCP scenarios (i.e., 15% in the historical period and approximately 25% in both future period scenarios). However, the spatial distribution patterns showed that RCP 4.5 was milder than RCP 8.5 for the very high values (i.e., ISImax > 75), with the exception of Corsica, where values greater than 75 appeared in an extended zone at the north of the island in RCP 4.5. Areas with values greater than 75 appeared in both scenarios in the central part of the study area (Var, Bouches-du-Rhône).

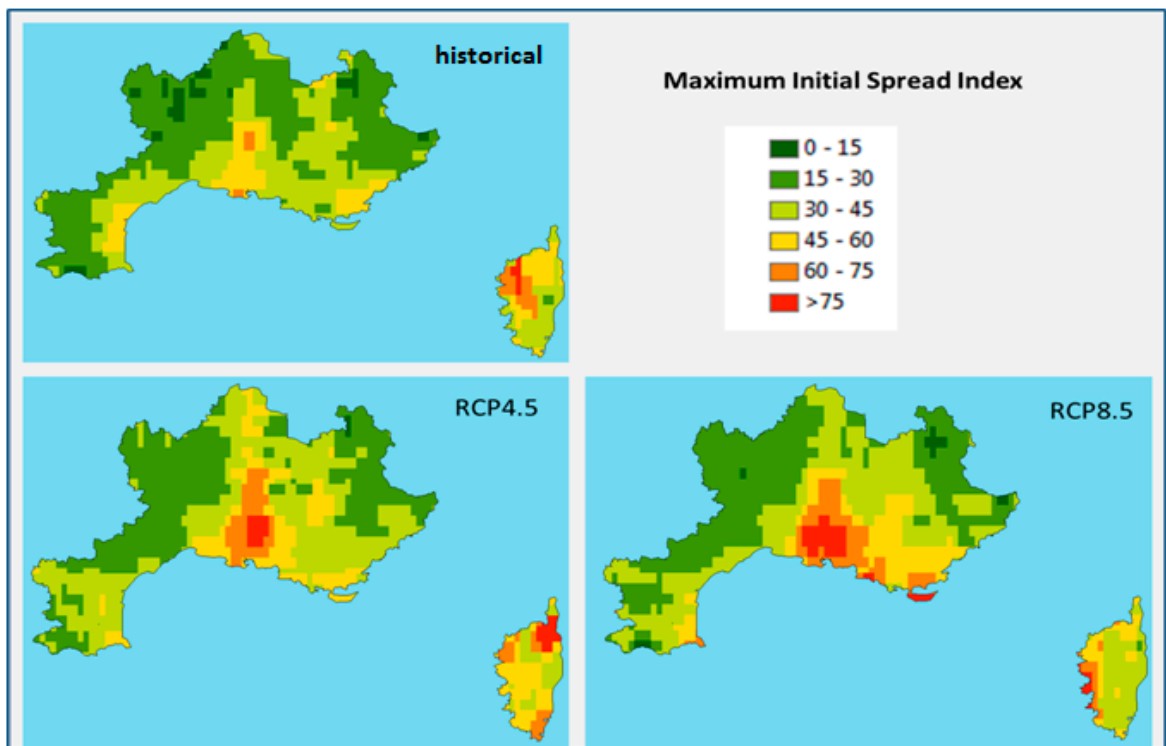

**Figure 5.** Maps of the historic (2006–2015) and near future (2036–2045) maximum values of Initial Spread Index for fire season.

**Table 1.** Distribution of areas in km$^2$ and area percentage changes in the categories of ISImax values (column 1), for the historical period (columns 2 and 3), RCP 4.5 (columns 4 and 5) and RCP 8.5 (columns 6 and 7).

| ISImax Values | Area (km$^2$)/Historical Period | Area %/Historical Period | Area (km$^2$)/RCP 4.5 | Area %/RCP 4.5 | Area (km$^2$)/RCP 8.5 | Area %/RCP 8.5 |
|---|---|---|---|---|---|---|
| 0–15 | 1188 | 3.464566929 | 72 | 0.209973753 | 378 | 1.102362205 |
| 15–30 | 16,560 | 48.29396325 | 12,618 | 36.79790026 | 12,834 | 37.42782152 |
| 30–45 | 11,376 | 33.17585302 | 13,068 | 38.11023622 | 12,654 | 36.90288714 |
| 45–60 | 4122 | 12.02099738 | 5652 | 16.48293963 | 4698 | 13.7007874 |
| 60–75 | 864 | **2.519685039** | 2124 | **6.194225722** | 2610 | **7.611548556** |
| >75 | 180 | **0.524934383** | 756 | **2.204724409** | 1116 | **3.254593176** |
| total | 34,290 | 100 | 34,290 | 100 | 34,290 | 100 |
| **>60** | **1044** | **3.03** | **2.880** | **8.21** | **3726** | **10.86** |

*3.3. Drought Mapping*

Drought Code mean values did not have significant changes for the future period for both climate scenarios (Table 2, Figure 6) According to the results presented in Table 2, the percentage of the area in the two higher categories of DC values (i.e., DC > 400) appeared with a mild change (i.e., about 2%) in the historical period and RCP 8.5, while there was a medium increase (about 5%) for RCP 4.5. These changes appeared mainly in the southern part of the area for RCP 4.5, while the spatial patterns of

Drought Code distribution were very similar in the three maps and particularly in the northern part and Corsica.

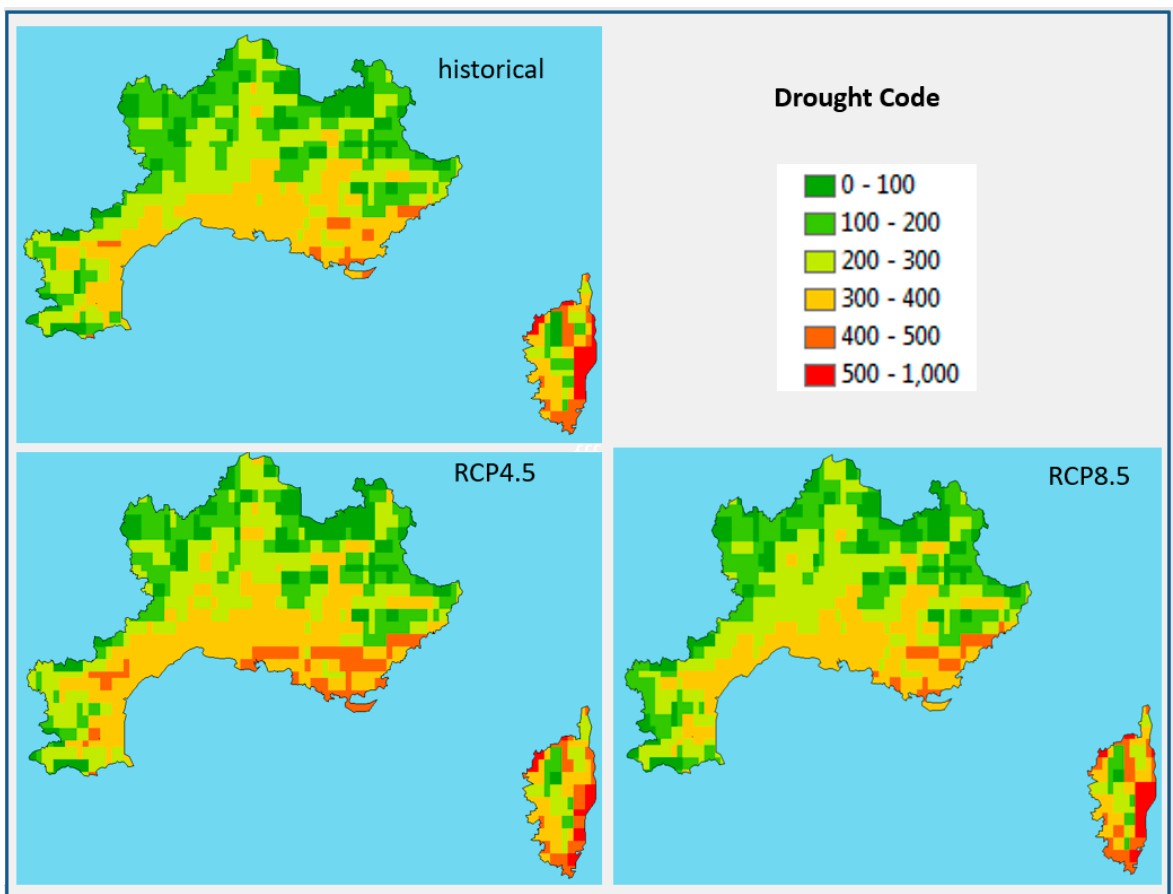

**Figure 6.** Maps of the historic (2006–2015) and near future (2036–2045) Drought Code mean values for fire season.

**Table 2.** Distribution of areas in km² and area percentage in the categories of DC mean values (column 1), for the historical period (Columns 2 and 3), RCP 4.5 (Columns 4 and 5) and RCP 8.5 (Column 6 and 7).

| DC Values | Area (km²)/Historical Period | Area %/Historical Period | Area (km²)/RCP 4.5 | Area %/RCP 4.5 | Area (km²)/RCP 8.5 | Area %/RCP 8.5 |
|---|---|---|---|---|---|---|
| 0–100 | 4500 | 13.12336 | 4302 | 12.54593 | 4446 | 12.96588 |
| 100–200 | 8316 | 24.25197 | 6534 | 19.05512 | 8280 | 24.14698 |
| 200–300 | 10,116 | 29.50131 | 9558 | 27.87402 | 10,512 | 30.65617 |
| 300–400 | 9342 | **27.24409** | 10,548 | **30.76115** | 8748 | **25.51181** |
| 400–500 | 1368 | **3.989501** | 2916 | **8.503937** | 1692 | **4.934383** |
| >500 | 648 | **1.889764** | 432 | **1.259843** | 612 | **1.784777** |
| total | 34,290 | 100 | 34,290 | 100 | 34,290 | 100 |
| **>400** | **2016** | **5.86** | **3348** | **9.75** | **2304** | **6.71** |

### 3.4. Number of Days with Extreme Fire Weather

According to Wang et al. [17], the number of days with FWI above a threshold reflect how extreme fire weather changes across space and time.

The resulting maps for both climate scenarios presented in Figure 7 and Table 3 indicate that there was a doubling on the number of days with very high and extreme fire weather danger (FWI > 50) per fire period in many locations within the study area and also that there was an important increase of the areas where FWI > 50 values occurred more than one week per fire period in the near future.

More particularly, the percentage of the area where FWI > 50 occurred more than seven days was found to increase from 5.8% in the historical period to 18.5% for the RCP 4.5 scenario and 10.6% for the RCP 8.5 scenario (Table 3) and especially for RCP 4.5; these sub-areas form a significant and compact "high danger" zone in the south-eastern part of the study area (see orange and red coloured area in Figure 7).

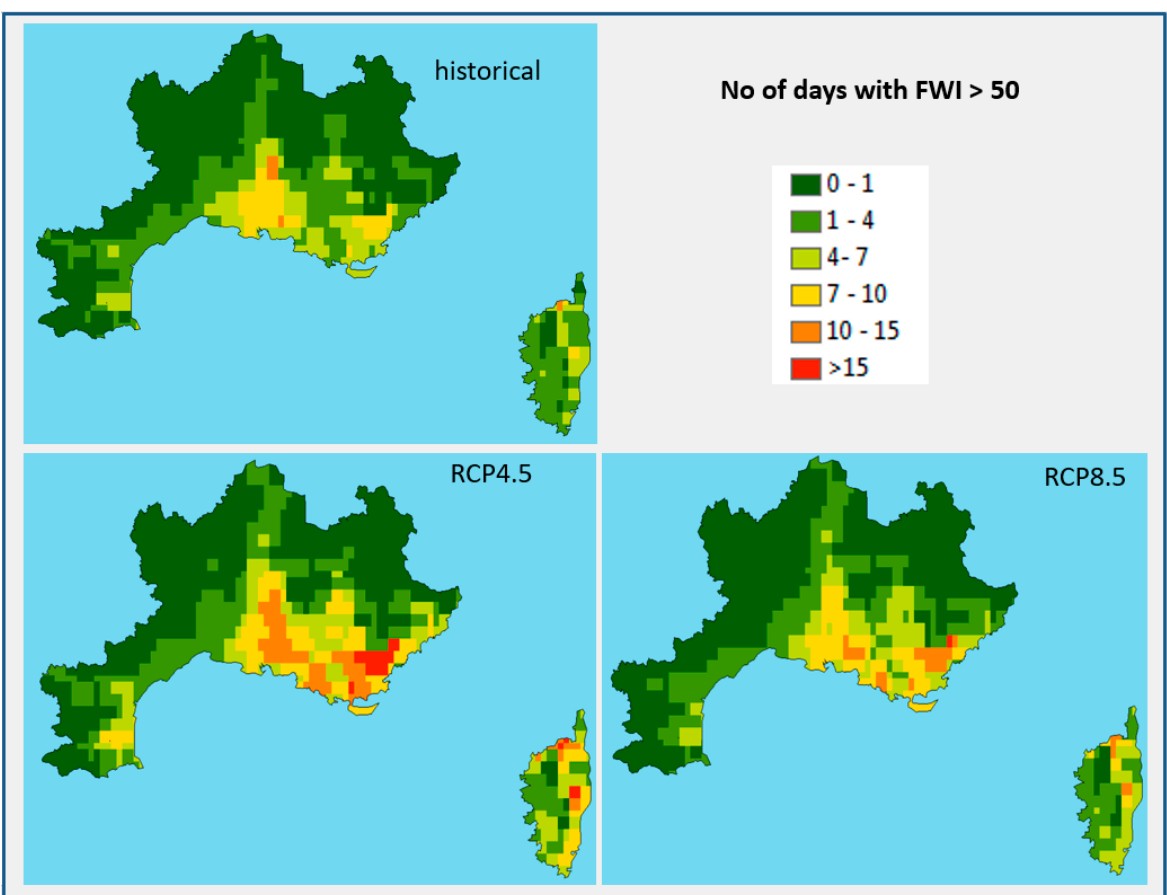

**Figure 7.** Maps of the historic (2006–2015) and near future (2036–2045) depicting number of days per yearly fire season.

**Table 3.** Distribution of areas in km² and area percentage in the categories of yearly number of days with extreme fire weather (FWI > 50) (Column 1), for the historical period (Columns 2 and 3), RCP 4.5 (Columns 4 and 5) and RCP 8.5 (Columns 6 and 7).

| Number of Days with FWI > 50 | Area (km²)/Historical Period | Area %/Historical Period | Area (km²)/RCP 4.5 | Area %/RCP 4.5 | Area (km²)/RCP 8.5 | Area %/RCP 8.5 |
|---|---|---|---|---|---|---|
| 0–1 | 18,270 | 53.2808399 | 15,300 | 44.61942257 | 17,550 | 51.18110236 |
| 2–4 | 9954 | 29.02887139 | 8442 | 24.61942257 | 8748 | 25.51181102 |
| 5–7 | 4068 | 11.86351706 | 4212 | 12.28346457 | 4356 | 12.70341207 |
| 8–10 | 1782 | 5.196850394 | 3672 | 10.70866142 | 2700 | 7.874015748 |
| 11–15 | 216 | 0.62992126 | 2052 | 5.984251969 | 900 | 2.624671916 |
| >15 | 0 | 0 | 612 | 1.784776903 | 36 | 0.104986877 |
| total | 34,290 | 100 | 34,290 | 100 | 34,290 | 100 |
| **>7** | **1998** | **5.826771654** | **6336** | **18.47769029** | **3636** | **10.60367454** |

## 4. Discussion

According to the above presented results, the extension of the forest fire risk area is evident in the near future using both RCP scenarios (4.5 or 8.5). The area with high to very high index values that raises concerns is increasing in the future and the various indexes show more severe conditions in all the examined cases. The only exception is the Drought Code parameter, which shows mild change

in the near future according to RCP 4.5 and no significant change for the RCP 8.5 (see Figure 6 and the last line of Table 2). The coastal sub-areas of PACA, Bouches-du-Rhône, Alpes Maritimes and Var, as well as the eastern part of Corsica are mostly affected. RCP 4.5 results in more severe conditions, for all the examined parameters, except of the maximum Initial Spread Index where RCP 8.5 provides a larger total area of extreme values (maxISI > 60). More particularly, the area that is affected by high FWI values for more than one week (see the last line of Table 3) in RCP 4.5 is almost double than in RCP 8.5 (Table 3). Regarding the spatial patterns of the higher categories of values, it appears that the affected areas are similar in both scenarios (i.e., southern–central areas of Provence-Alpes-Côte d'Azur (PACA) area and the coastal areas of Corsica); however, the changes are more dramatic in RCP 4.5, where the very-high and extreme categories of values are larger. The explanation for this is that the simulated climatic data, which are the inputs of the FWI system, are more favourable to higher fire danger estimation in the scenario RCP 4.5, which is related to difference in the distribution of emissions and the generated heat from the models. An analysis of the climatic data for the climatic models that were used in this study and for each RCP scenario can provide more detailed information on the comparison of the input data derived from the RCPs; however, this is beyond the scope of this research. Different spatial patterns depicting maxISI higher classes in Corsica appear in the historic and future estimations (i.e., central western part in historic maxISI, northern part in RCP 4.5, eastern part in RCP 8.5). This is a notable finding, as it does not allow the extraction of conclusions for the expected spatial changes of this index for Corsica.

This present work relates to the impact of climate change on fire danger in the near future horizon in the study area based on climate model simulations of the two RCP scenarios, which are built on assumptions of socioeconomic and land-use change projections. Therefore, the presented results reflect future potential fire danger and risk changes that can be the basis for the design of management actions for fire prevention planning and mitigation of forest fire impacts in the area of interest.

At this point, the acquired results from the present study clearly indicate that climate change will affect this part of France in the near future horizon and this has to be taken into account for the improvement and appropriate adaptation of the prevention plans and also of the optimization of the operational national doctrine. In future works, the aim will focus on the use of two RCM models forced by a number of GCM model outputs in order to assess the bias patterns and climate change signals more efficiently. Additional indexes for drought will be studied as well.

## 5. Conclusions

According to the results of the current study, various levels of changes in fire danger are expected in the near future due to climate change in the study area. The indices that depict significant changes are those related to fire suppression effort and difficulty, namely the Initial Spread Index, the Seasonal Severity Rating and Fire Weather Index (ISI, SSR and FWI, respectively). The number of days with high and very high FWI values were found to double in particular areas of the PACA region and Corsica. The areas with high Initial Spread Index and Seasonal Spread Index values were found to increase as well, forming compact zones of high values in the southern part of the study area, while the changes in the Drought Code were negligible. The current study on the evolution of spatial and temporal distribution of Forest Fire Danger due to climate change can provide important knowledge to the design of the near future forest fire management actions at a national level and also the compilation of guidelines for the adaptation of forest fire related policies at national and EU levels.

**Author Contributions:** Conceptualization, V.V. and D.V.; methodology, V.V. and A.S.; software, V.V. and S.K.; validation, T.S. and F.G.; formal analysis, V.V., D.V., A.S. and N.P.; investigation, V.V., D.V. and A.S.; resources, V.V., D.V., A.S. and S.K.; data curation, V.V., N.P. and S.K.; writing—original draft preparation, V.V.; writing—review and editing, D.V. and F.G.; visualization, V.V.; supervision, D.V.; project administration, A.S.

**Funding:** This research was funded by the European Union's Horizon 2020 research and innovation programme under grant agreement No 653824 (EU-CIRCLE) and the APC was funded by NCSR "Demokritos".

**Acknowledgments:** The authors would like to thank the climate modelling groups of CNRM-CERFACS, MPI-M-MPI-ESM-LR and SMHI-RCA4 for making available the data sets.

**Conflicts of Interest:** The authors declare no conflict of interest. The funders had no role in the design of the study; in the collection, analyses, or interpretation of data; in the writing of the manuscript; or in the decision to publish the results.

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
