# Peer review of "Projection of Forest Fire Danger due to Climate Change in the French Mediterranean Region"

_sustainability, doi:10.3390/su11164284_

Reviewer 1 Report

Dear Autors,

Your article is very interesting from the point of view of climate change and the danger of forest fire. I would congratulate for the idea and interest on this topic.

Suggestions for improve the paper:

Abstract: a huge part of abstract is related to method not to the results. Also the importance of study can be highlighted.

Introduction: this chapter is too short. Also, some new research can be inserted and compare the results. We understand that is an projection but the base (fundamentals) can be improve.

Author mention that FWI system was use to for some studies but some discussion can be made. Also, the limitation is not presented.

This region is important from the point of view of fire? What is the dimension of these problems here? What are the real cause of these forest fire here? What are the structure of vegetation and the measure to reduce these impact? 

Materials and method: 

line 101: formula is your own calculation or is baase on a reference?

The sub-chapter description of the study area and Climate models and Methodology can be re-numbering-2.1, 2.2, 2.3.

For me is not very clear in this chapter what are the type of data use in this projection? Only Euro-CORDEX? This data is representative?

I suggest that this chapter must be improve. More clear and data use.

In the paper is not mention why the autor use only: DC, ISI, FWI, and SSR.

Results.

This part must be improve. We suggest to use some statistical analysis for relevance of scenarios. Also, some correlation with present real situation must be done. Only graphic results and no statistical analysis.

Discussion.

These Chapter must be improve. Is very short.

For example what are the effort to reduce these impact in the region? Is some support from authorities? For the protected areas is some measures in the management plans? 

Conclusions is missing.

The chapter References must be corrected. Is some mistake in formating and arrange of references.

Reviewer 2 Report

The spatial and temporal distribution of Forest Fire Danger related to climate change is one of the most important topics for fire science. In this paper the authors address the topic following a correct theoretical approach. However, in addition to the need to take more care of the editing part, my main observation concerns the incompleteness of the article from a methodological point of view. In particular the critical points in my opinion are the following:

1. the authors do not adequately describe, above all from the quantitative point of view, the way in which they use the models and the climatic scenarios to estimate the future meteorological parameters required for the calculation of the Fire Weather Index components (Row 185-192);

2.  the choice to show the results of only three FWI parameters is not explained;

3. A calculation methodology of three FWI parameters is not reported;

4. Climate models and scenarios are roughly described.

Reviewer 3 Report

The technical jargon used in the text is perhaps only suitable for experts in the use of computer tools used by authors. The results seem like a mere exercise in applying some assumptions to see what comes up.

The figures are little explained and in the case of figure 1 the text seems not to match the represented scheme.

The chosen scenarios are not clearly explained and the reader can deduce that the final result will depend on how correct or erroneous the starting hypotheses are.

Author Response

Round  2

Reviewer 1 Report

Dear Autors,

- congratulations for improvement on this manuscript.

Minor changes to be consider:

-please check format editing and some error of tyyping.

-chapter discussion can be more developed.

-Table 1, 2 and  is introduced. For this table is not very clear what means and what are formula for calculation (tis must be explained)- ex.  Area % - /RCP 4.5. Also, more discussion related to this data can be made.

-some statistical differences bethween scenarios can be made. Base on this differences some new discussion can be made (what scenario is more relevant, why in the 4.5 scenario all data are lower and what are the possible explanation for this data in the case of all index.

Reviewer 2 Report

I think the answers to my previous comments are not enough. In my opinion this work is weak in terms of the mathematical approach which is however necessary in the specific case.
My questions are as follows:
How is
calculated DC? Based on which equation? How are the necessary parameters obtained?
How is
calculated ISI  which depends on FFMC (which therefore must be calculated) as well as on the wind?
How is
calculated FWI  which depends on ISI but also on BUI (which should then be calculated)?

Reviewer 3 Report

The text has been significantly improved

Author Response

We tried to clarify more some points in the results and also the titles of the tables.

Thank you very much for your review!

Round  3

Reviewer 2 Report

Unfortunately, compared to the points indicated in the previous review, I am forced to consider the answers insufficient. In the specific case it is obviously not a question of inserting the formulation of the entire algorithm in the paper, but of answering in a more adequate manner to the questions.
